# Controlled dissolution of a single ion from a salt interface

Huijun Han [1], Yunjae Park [2], Yohan Kim [1], Feng Ding [1,3] ✉ & Hyung-Joon Shin [1,2] ✉

Interactions between monatomic ions and water molecules are fundamental to understanding the hydration of complex polyatomic ions and ionic process. Among the simplest and well-established ion-related reactions is dissolution of salt in water, which is an endothermic process requiring an increase in entropy. Extensive efforts have been made to date; however, most studies at single-ion level have been limited to theoretical approaches. Here, we demonstrate the salt dissolution process by manipulating a single water molecule at an under-coordinated site of a sodium chloride film. Manipulation of molecule in a controlled manner enables us to understand ion–water interaction as well as dynamics of water molecules at NaCl interfaces, which are responsible for the selective dissolution of anions. The water dipole polarizes the anion in the NaCl ionic crystal, resulting in strong anion–water interaction and weakening of the ionic bonds. Our results provide insights into a simple but important elementary step of the single-ion chemistry, which may be useful in ion-related sciences and technologies.

Ions, the building blocks of materials, interact with counter-ions or polar molecules through long-range electrostatic forces, which can break even ionic bonds. Applications for ions have been found in various fields, including electrochemistry, catalysis, battery, and biology[1–5]. Despite the great importance of ionic processes, however, the intrinsic features of single ion have rarely been examined experimentally, because the dissolution process occurs in solution. Most previous experiments have measured only the averaged features of ions[6–8]. When sodium chloride (NaCl) dissolves in water, the positive enthalpy change competes with the entropy increase, so the reaction is influenced by both temperature and composition[9,10]. In the microscopic aspects, none of the experimental results can determine which ion is dissolved first or why splitting of ionic bonds is initiated by weak ion–water interactions[11–17]. The dissolution process has been substantially addressed in theoretical studies and its origin was debated in early works[18–25]. It was reported that strongly bound water molecules attract or repel the anion from the salt crystal[18–20]. A recent study suggested that the polarizability difference between cations and anions increases the

entropic contribution to the system[23]. However, experimental evidence of theoretical predictions is still lacking; and thus, direct observation of the dissolution process at the single-ion level has long been required.

Herein, we report the real-space observation of selective dissolution of salt by manipulating a single water molecule. Selective dissolution can be attributed to the difference in ion–water interactions between anions and cations. The water molecules were investigated on ultrathin NaCl(100) films supported on single-crystal Ag(100) as a model system by means of scanning tunnelling microscopy (STM) and density functional theory (DFT) calculation. We examined the dynamics of single water molecules along with the adsorption structures on the surface. We successfully controlled the molecule through lateral manipulation, thereby elucidating the cation–water and anion–water interactions. Specifically, the manipulation at a step of a NaCl film makes a single Cl⁻ ion released out of the step, creating a Cl⁻ vacancy. The anion with higher polarizability than the cation is polarized by the dipole of the water molecule, which weakens the ionic bonding and results in selective dissolution in the initial stage.

[1]Department of Materials Science and Engineering, Ulsan National Institute of Science and Engineering (UNIST), Ulsan 44919, Republic of Korea. [2]Center for Multidimensional Carbon Materials, Institute for Basic Science (IBS), Ulsan 44919, Republic of Korea. [3]Shenzhen Institute of Advanced Technology, Chinese Academy of Science, Shenzhen 518055, China. ✉e-mail: f.ding@siat.ac.cn; shinhj@unist.ac.kr

## Results and discussion

### A model system for salt dissolution

Two-monolayer (2 ML) NaCl(100) film supported on Ag(100) surface was partially covered with one additional layer (3 ML NaCl). We selected the 2 ML/3 ML NaCl interface as a model system for the dissolution experiment because the dissolution process usually begins at atomic steps. Water molecules were dosed at temperatures below 20 K and hence existed as individual entities (Fig. 1a). The water molecules, which inhabited both the interfacial steps and the terrace, exhibited similar adsorption behaviors in the two local environments. To discern the adsorption site of a single $H_2O$ molecule, high-resolution images were obtained at both environments. Only $Cl^-$ ions were imaged as protrusions in STM images due to its high density of state near the Fermi level. In Fig. 1b, a single water molecule on the terrace was located at the hollow sites of $Cl^-$ lattice (white dash)[26]. Similarly, the molecule at the non-polar step also adsorbed at a position between two $Cl^-$ ions, i.e., a $Na^+$ site (Fig. 1e). The preferential adsorption near $Na^+$ ions originated from an electrostatic interaction between the cation and two lone pairs of electrons of the oxygen atom. The water molecule at step is thought to be strongly attracted by two adjacent $Na^+$ ions, one from the step and the other from the terrace.

With the DFT approach, we obtained the optimized structures and the energies of an isolated water molecule on the terrace and at the step, respectively (Fig. 1c, d, f, g). On the terrace, the molecular plane of water is nearly parallel to the NaCl surface. The oxygen atom of the water molecule sits at $Na^+$ ion, with two OH bonds pointing towards two $Cl^-$ ions. At the step, the oxygen atom of the water molecule resides in the vicinity of two $Na^+$ ions, one on the lower terrace and the other at the step. One hydrogen atom points towards a $Cl^-$ ion of the step and the other one away from the surface. The adsorption energies ($E_{ads}$) of the water on terrace and near step edge are −385 and −750 meV, respectively, which proves that the interfacial step is energetically highly preferred for the adsorption of the water molecule.

### Interaction of a water molecule with anions and cations

The dissolution of NaCl was achieved by the lateral manipulation of the single molecule at the interfacial steps. Prior to dissolving the step, we need to control the individual molecules on the terrace of the NaCl films. The molecules were laterally dragged with an STM tip at a sample bias lower than 450 mV (OH stretching mode of water molecule) to avoid undesirable hopping or desorption[27]. At a closer tip–molecule distance of about 300 pm, we moved the tip along the non-polar <100> and polar <110> directions on the terrace, as illustrated in Fig. 2a–d[28]. Figure 2e shows the corresponding tip traces followed by

the molecules and the atomic corrugations of NaCl surface along both directions. The average spacings between the dips ($Na^+$ sites) along the non-polar and polar directions of the tip traces were 0.556 and 0.387 nm, respectively. These spacings were consistent with the distances of $Na^+$–$Cl^-$–$Na^+$ (0.564 nm) and $Na^+$–$Na^+$ (0.399 nm), respectively. The manipulation was controlled by the tip–molecule attraction and was not dominated by the electric field because the molecules followed the tip regardless of the polarity and the magnitude of bias voltage (Supplementary Fig. 1)[29].

Although the molecule was manipulated under the same conditions, the amplitude of its propagation varied with the manipulation direction (Fig. 2e). The average height of the modulation was lower along the non-polar direction (6.68 pm) than along the polar direction (10.88 pm). In contrast with these modulations, the atomic corrugation on the NaCl(100) surface was higher along the non-polar direction (3.05 pm) than along the polar direction (1.84 pm) in Fig. 2e, f, which means that the polar direction is smoother than the non-polar direction. The modulation along the tip trace includes both the surface corrugation and the molecule–substrate interaction. These results indicate that, along the non-polar direction, the molecule–substrate distance over the $Cl^-$ site should be reduced. The tip height traces obtained using different tips with various conditions also yielded the comparable result (Supplementary Fig. 1a). The anion–water interaction is possibly accompanied during the lateral manipulation, by molecular rotation that aligns the dipole of the molecule towards the anion (Fig. 3a, b). The dipole moment of water polarizes the anion, increasing the attractive anion–water interaction. This phenomenon would explain the reduced amplitude of modulation in dragging along the non-polar direction. As the $Cl^-$ anion is 20 times more polarizable than the $Na^+$ cation[30], its electron cloud is easily concentrated towards the dipole moment of the molecule. The trace of the water molecule thus revealed that the different height of corrugations is associated with molecule–substrate interactions.

### Manipulation-induced anion–dipole interaction

The periodic array of atoms of the substrate yields an equal displacement for each movement in the tip-height trace (Fig. 2e). A sudden jump in the trace implies that the water molecule is pulled by the tip and hops by one adsorption site toward the tip. The lateral motion of a water molecule is initiated when the lateral component ($F_L$) of the tip–molecule attractive force ($F_T$) exceeds the threshold frictional force between a molecule and a substrate ($F_{TH}$)[31]. At the moment of jump, the lateral force reaches the threshold frictional force, and the angle ($\phi$) between $F_T$ and $F_L$ can be obtained from the slope of the tip

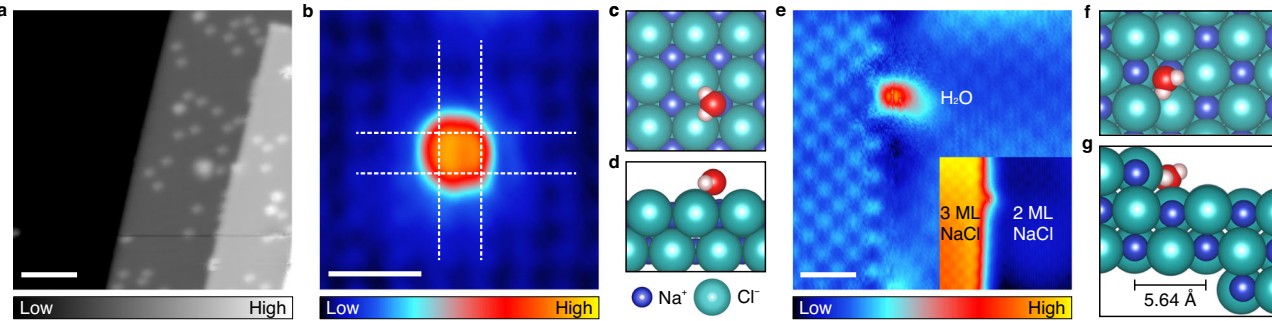

**Fig. 1 | Water molecules on a NaCl surface. a** STM image of water molecules on 2 and 3 ML NaCl surfaces ($V_s$ = 200 mV; sample bias and $I_t$ = 50 pA; tunnelling current). Scale bar represents 6 nm. **b** High-resolution STM image of a single water molecule on the 2 ML NaCl surface detected with a water-terminated STM tip ($V_s$ = −200 mV, $I_t$ = 50 pA). Dotted lines represent $Cl^-$ lattice. **c, d** Top and side views of the optimized configuration of water molecule on the terrace calculated by DFT.

**e** Flattened high-resolution STM image of a single water molecule at the step ($V_s$ = 300 mV, $I_t$ = 500 pA). The inset shows an unprocessed image. Scale bars in **b** and **e** represent 1 nm. **f, g** Top and side views of the optimized configuration of water molecule at the step calculated by DFT. In **c, d, f** and **g**, blue, blue-green, red, and white spheres represent $Na^+$ ions, $Cl^-$ ions, oxygen atoms and hydrogen atoms, respectively.

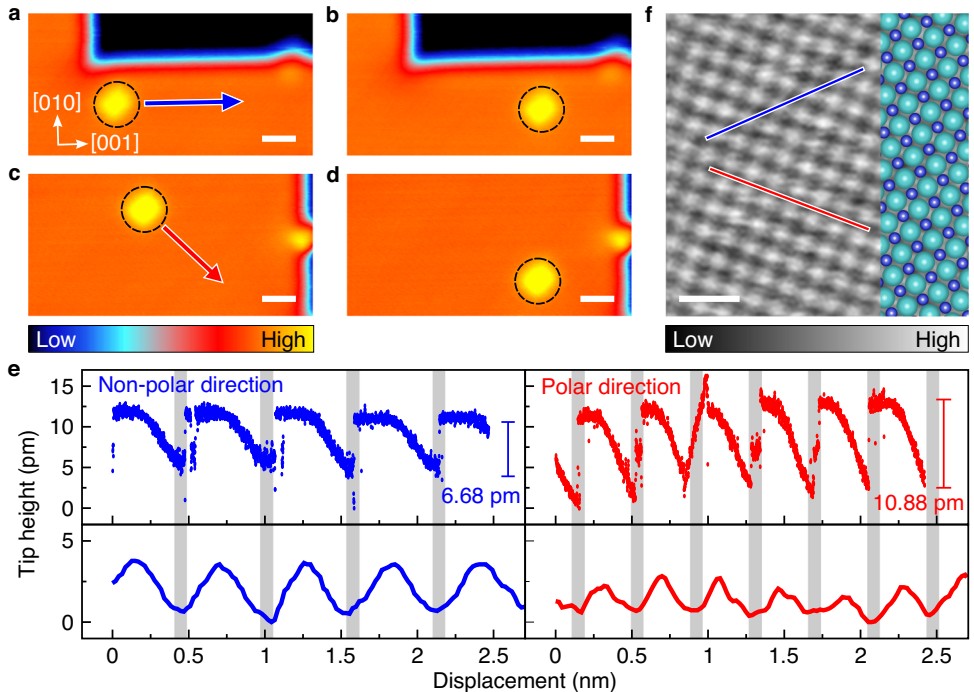

**Fig. 2 | Lateral manipulation of a single water molecule on the terrace. a–d** STM images of a single water molecule on the 3 ML terrace before and after a lateral manipulation along (**a**, **b**) the non-polar direction (blue arrow) and (**c**, **d**) the polar direction (red arrow), respectively ($V_s$ = 200 mV, $I_t$ = 50 pA). The molecules were manipulated under $V_s$ = 200 mV and $I_t$ = 800 pA at a speed of 50 pm s$^{-1}$. **e** Tip-height traces of the molecules on the terraces (blue dots; non-polar direction, red dots;

polar direction) and surface corrugations (blue line; non-polar direction and red line; polar direction) from **f**. The shaded region indicates the position of Na$^+$ ion. **f** High-resolution STM images of the 3 ML NaCl terrace ($V_s$ = 200 mV, $I_t$ = 200 pA). Scale bars in **a–d** and **f** represent 1 nm. Blue and red lines are the surface corrugations along non-polar and polar directions in **e**, respectively. Blue and blue-green spheres are superimposed to represent Na$^+$ ions and Cl$^-$ ions, respectively.

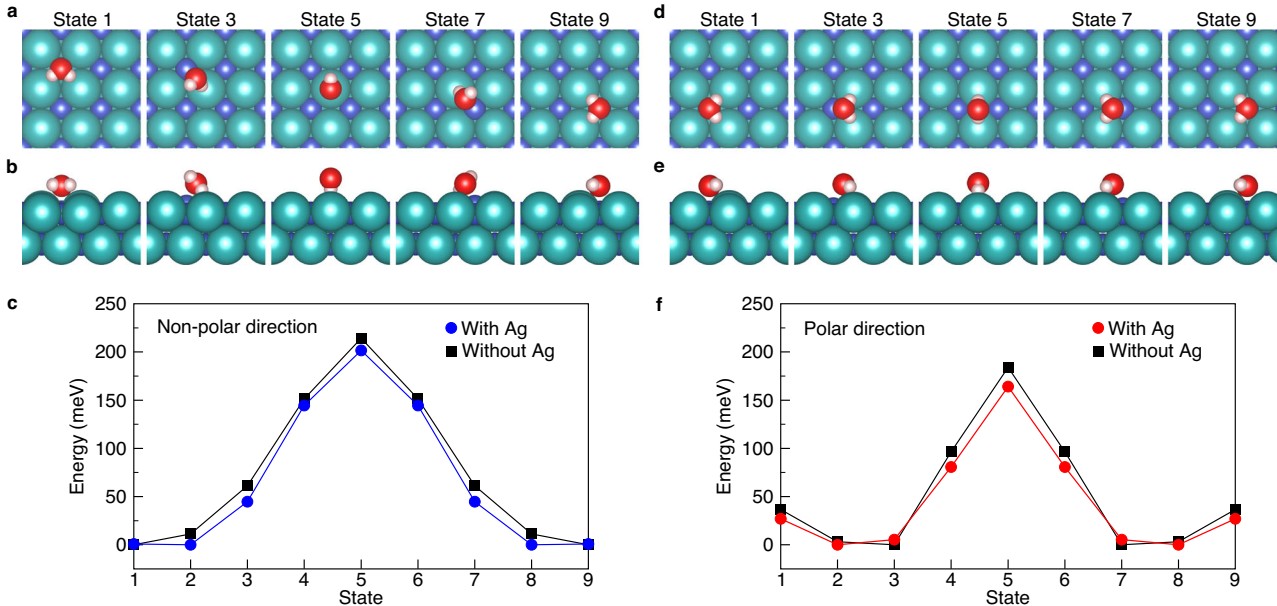

**Fig. 3 | Calculated energy profile of the lateral manipulation of a water molecule on 2 ML NaCl surface. a**, **b** Top and side views of the optimized configurations of water molecule along the non-polar direction on a 2 ML NaCl. **c** Energy profiles of water molecule moving along the non-polar direction on the 2 ML NaCl with and without Ag(100) substrate. **d**, **e** Top and side views of the optimized configurations

of water molecule along the polar direction on a 2 ML NaCl. **f** Energy profiles of water molecule moving along the polar direction on the 2 ML NaCl with and without Ag(100) substrate. The optimized configurations of a water molecule on a 2 ML NaCl/Ag(100) along both directions were identical with above configurations.

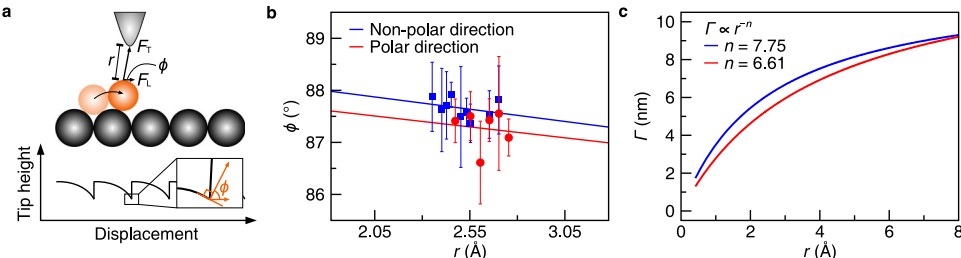

**Fig. 4 | Manipulation of the molecule with different heights. a** Schematic of dragging adsorbate on a surface. Tip-adsorbate distance ($r$), total force ($F_T$), its lateral component ($F_L$) and angle ($\phi$) between $F_T$ and $F_L$ are given with a typical tip trace. **b** The $\phi$ versus $r$ plot. Blue and red points indicate that the molecule was manipulated along the non-polar and polar directions, respectively. Each point was averaged over at least six trials. The error bars indicate the standard deviation. The solid lines are the least-squares fitted lines of the data. As the tip approached the molecule, the required angle for manipulating the molecule increased to compensate the increased lateral component of the tip–molecule force. **c** The $\Gamma$ ($= r/\cos(\phi)$) versus $r$ plot derived from the fitted lines in **b**. Blue and red lines indicate that the molecule was manipulated along non-polar and polar directions, respectively.

trace (Fig. 4a). The $F_{TH}$ is then described as[31]

$$F_{TH} = F_T \cos(\phi). \tag{1}$$

The manipulation along the non-polar direction revealed the higher angle than along the polar direction (Fig. 2 and Supplementary Fig. 1b), indicating lower threshold frictional force. The non-polar array is an easy direction of motion owing to the anion–dipole interactions.

To further investigate the anion–dipole interaction, we measured the tip traces at different tip–molecule distances ($r$) along both directions. When the tip–molecule distance decreases, the angle increases with the strengthened tip–molecule interaction. In terms of the tip–molecule interaction ($U$), Eq. (1) can be rewritten as

$$F_T = \frac{U}{r} = F_{TH} \frac{1}{\cos(\phi)} = F_{TH} \frac{\Gamma}{r} \tag{2}$$

where $\Gamma = r/\cos(\phi)$[32]. The $\Gamma$ plot describes the attractive part of the interatomic potential with a coefficient, $F_{TH}$, but requires the exact value of $r$. The tunnelling gap between the tip and the Ag substrate ($z_{Ag}$) was extracted from $I$–$z$ spectroscopy. We can estimate the distance between the tip and the molecule on the NaCl surface (see Supplementary Fig. 2 and Methods). Figure 4b shows the changes in the angle as a function of tip–molecule distance. The angle clearly increased with decreasing $r$, implying that $U$ increased (Supplementary Fig. 3). Inserting the fitted lines in Fig. 4b into Eq. (2), we found a negative power-law relation between $\Gamma$ and $r$, $\Gamma \propto r^{-n}$ with $n = 7.75$ and 6.61 in the non-polar and polar directions, respectively (Fig. 4c). The larger exponent along the non-polar direction reflects that the molecule strongly interacts with the tip, also implying higher energy barrier. Although the water molecule has a higher energy barrier along the non-polar direction (Fig. 3), the experimental results exhibit lower frictional force along this direction (see Supplementary Fig. 4 and "Methods" section). According to a sinusoidal model, the surface potential is given by $V(x) = \frac{1}{2} E_b \cos(2\pi x/a_0)$, where $E_b$ and $a_0$ are the energy barrier and the spacing between the nearest Na$^+$ sites[33]. Threshold frictional force ($F_{TH} = (dV/dx)|_{max}$) calculated using the DFT results (Fig. 3) shows that $F_{TH}$ becomes lower along the non-polar direction than the polar direction, which agrees well with experimental results. Therefore, the water molecules adsorb on the cations but interact strongly with the anions during the manipulation due to the anion–dipole interaction.

### Selective dissolution of anion by polarization

When manipulating a water molecule along the step (Supplementary Fig. 5), we sometimes observed structural changes of the straight step. In Fig. 5a, two water molecules occupied the step and the molecule at the right was dragged leftwards. After this manipulation, a depression

was created at the step site, and a bright spot was also created on the step (Fig. 5b). The depression was a vacancy at the step site because the molecule remained intact without dissociation during the manipulations (Supplementary Fig. 6). The ion, extracted by the water manipulation, was re-adsorbed near the vacancy (Fig. 5b). Figure 5c shows a single Cl$^-$ vacancy at the step. The Cl$^-$ ions adjacent to the vacancy were also slightly shifted from their original positions, implying that the anion was indeed extracted[34]. Figure 5d shows another NaCl step from which a Cl$^-$ ion was extracted by manipulation (Supplementary Fig. 7). In this case, the hydrated species was not found on the surface. The symmetric feature of the defect revealed that two Cl$^-$ ions near the vacancy were displaced towards the upper terrace and their heights were lowered due to the reconstruction of the lattice by the missing Cl$^-$ ion. The simulated STM images also support the extraction of a single Cl$^-$ ion (Supplementary Fig. 8). This lateral manipulation of a single H$_2$O molecule at a NaCl step leads to preferential dissolution of a single Cl$^-$ ion over a single Na$^+$ ion. In contrast, single H$_2$O molecules preferentially adsorb at Na$^+$ sites over Cl$^-$ sites. In other words, the cation–molecule interaction is much stronger than the anion–molecule interaction, raising the question: why does water molecule dissolve only chlorine ion?

The polarizability ($\alpha$) plays a crucial role in the selectivity of the dissolution process[16,22,23,35,36]. The polarizabilities of the anion and cation are greatly discrepant[30]: $\alpha$(Cl$^-$) = 3.5 Å$^3$; $\alpha$(Na$^+$) = 0.18 Å$^3$. The total interaction of an alkali halide crystal can be divided into interatomic, Coulombic, and polarizable interactions. The polarizable interaction contributes minorly to the total interaction in ionic bonding, but is crucial in the dissolution process. In NaCl, the anion has a flexible electron cloud[37] and the charge of the cation is tightly bound to the nucleus[38]. The electron cloud of the anion is concentrated towards the cation, and hence the charge of the anion by the polarizable interaction becomes redistributed against the electron transfer from Na atom to Cl atom. The polarization of the anion decreases the Coulombic interaction between the two ions. This interaction is trivial for ions in bulk because the coordinating ions cancel out the polarization effect. However, the polarizable interaction becomes significant at defect sites, especially at steps and kinks where the symmetry is broken. As the polarizable interaction cannot be eliminated at defect sites, dissolution takes place there first.

Owing to polarizability, a single Cl$^-$ ion will more likely be extracted than a single Na$^+$ ion or a Na–Cl ion pair. Figure 5e–h shows the charge density difference plots and its cross-sectional plots of a water molecule at the interfacial step. The water molecule adsorbed near Na$^+$ ion (Fig. 5e, f and the state 3 in Supplementary Fig. 5c) does not affect the charge density at Na–Cl bond of 3 ML NaCl. At the interfacial Cl$^-$ site (Fig. 5g, h and the state 11 in Supplementary Fig. 5c), however, the charge is concentrated to the water molecule and is depleted at the Na–Cl bond. The water molecule at the Cl$^-$ site weakens

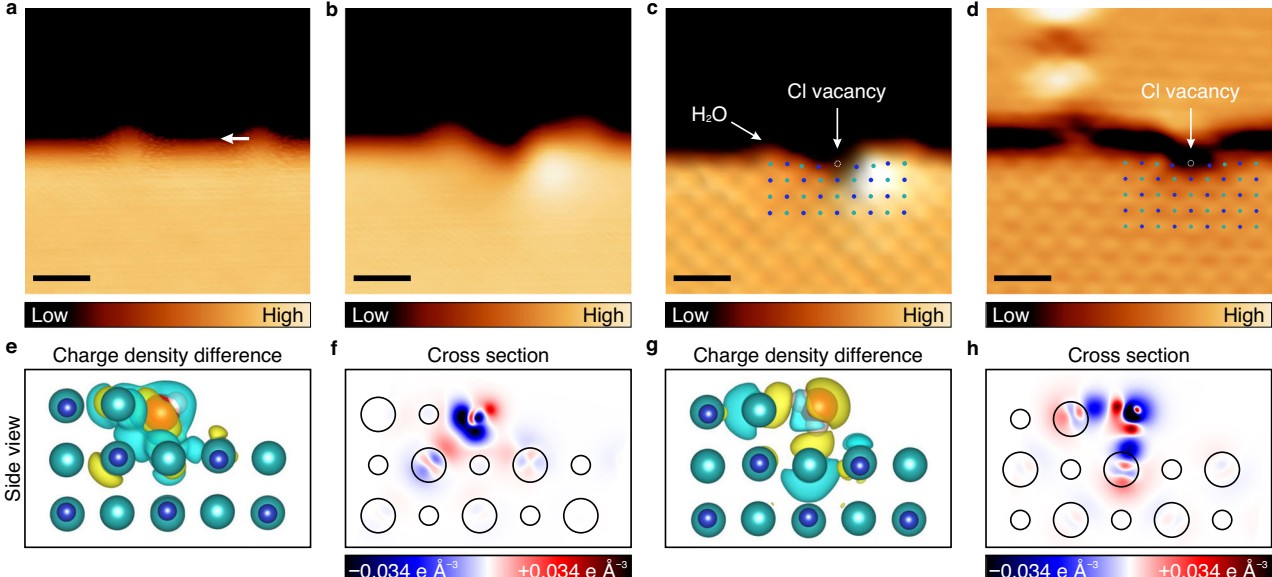

**Fig. 5 | Selective dissolution of a single Cl⁻ ion from the step. a, b** STM images of water molecules at the step edge before and after manipulation along a white arrow, respectively ($V_s$ = 300 mV, $I_t$ = 50 pA). **c** High-resolution STM image of **b** showing a single Cl⁻ vacancy ($V_s$ = 300 mV, $I_t$ = 500 pA). **d** Flattened STM image of another example of the selective dissolution in Supplementary Fig. 7j, clarifying displaced ions near the vacancy ($V_s$ = 400 mV, $I_t$ = 500 pA). Scale bars in **a**–**d** represent 1 nm. In **c** and **d**, blue and blue-green dots indicate Na⁺ and Cl⁻ ions and dotted circles represent vacant Cl⁻ sites of the steps. **e**–**h** Side view of charge density difference and cross-sectional plots of the water molecule at the Na⁺ site (**e**, **f**) and at the Cl⁻ site of the step (**g**, **h**) calculated by DFT. Blue, blue-green, red, and white spheres represent Na⁺ ions, Cl⁻ ions, oxygen atoms and hydrogen atoms, respectively. In **e** and **g**, yellow (cyan) indicates an area of electron accumulation (electron depletion). In **f** and **h**, large (small) circle indicates the position of Cl⁻ ion (Na⁺ ion).

the ionic bonds by the molecular rotation during translation. When a water molecule rotates during translation, one OH bond points to the Cl⁻ ion, distorting and attracting the electron cloud of the anion through the anion−dipole polarizable interaction. This distortion leads to the depletion in a charge population between the cation and the anion, which decreases the Coulombic nature of the ionic bonding. The electron cloud of the Cl⁻ ion, which contributes to Coulombic interactions, is concentrated at the H₂O molecule, in turn weakening the ionic bonds of NaCl. Therefore, aided by the H₂O molecule, the Cl⁻ ion was selectively extracted. This reaction is thermodynamically unfavorable without the entropic effect, but our result shows the significance of the polarizable interactions.

This microscopic dissolution process demonstrated by single-molecule manipulation reflects the real dissolution process under ambient conditions, where numerous water molecules dissolve the salt spontaneously[39,40]. In such crowded conditions, the adsorption of multiple water molecules can severely distort the electron cloud of a Cl⁻ ion compared to a single water molecule, thereby facilitating easier dissolution and hydration of the Cl⁻ ion. Once a Na⁺ kink is created, which offers additional adsorption sites with lower coordination number, ionic bonds become weaker, promoting the dissolution of ions[41,42]. Eventually, the sequential dissolution proceeds as kinks propagate until the solubility limit is reached.

We investigated the selective dissolution of single Cl⁻ ion at a monoatomic step on a NaCl crystal. The ion−H₂O interaction was examined by laterally manipulating a single H₂O molecule along the polar and non-polar directions on the terrace. In contrast to isolated H₂O molecules, which always adsorb at the Na⁺ sites on both terraces and steps, the molecule strongly interacted with the Cl⁻ ion during the lateral manipulation, due to the polarizable interaction. As a result, the H₂O molecule in the present manipulation experiment extracted a single Cl⁻ ion, leaving a single Cl⁻ vacancy. As it traveled, the H₂O molecule rotated to align its dipole with the surface ions. This rotation maximized the H₂O−Cl interaction by dipole-induced dipole interaction and decreased the ionic bonds in NaCl. The

selective dissolution of chlorine anion is attributed to its large polarizability.

## Methods
### STM measurement
Experiments were carried out in an ultrahigh-vacuum low-temperature scanning tunnelling microscopy (SPECS Joule-Thomson STM, JT−STM) at 4.4 K. The base pressure was of the order of $10^{-11}$ Torr. The Ag(100) single-crystal substrate (Mateck, 99.999%) was cleaned by several cycles of sputtering and annealing, then deposited with the NaCl film by thermal evaporation of the NaCl source (Sigma Aldrich, 99.999%). During the deposition process, the substrate temperature was maintained between 333 and 343 K to build 2 ML island and an additional NaCl layer on the 2 ML island. The water molecules (Sigma Aldrich, deionized) were purified through several cycles of the freeze-thaw method and were dosed by exposing the substrate to water vapor at a temperature below 20 K. As the probe, we used an electrochemically etched tungsten tip cleaned by several repeats of high bias pulses and tip-crash technique on the Ag substrate.

The initial tip height was set at a tunnelling condition of 350 mV and 50 pA. We estimated the tip height on the Ag substrate, $z_{Ag}$, to be 786 pm by measuring $I$-$z$ spectroscopy (Supplementary Fig. 2a). Combining the apparent height ($\Delta z_{NaCl}$, 450 pm) with the physical dimension ($d_{NaCl}$, 846 pm) of the 3 ML NaCl, we obtained the tip height on the film ($z_{NaCl}$, 390 pm) as follows:

$$z_{NaCl} = z_{Ag} + \Delta z_{NaCl} - d_{NaCl}. \tag{3}$$

We estimated the tip-molecule distance ($r$) using a simple geometric relation in Fig. 4a; $r = z/\sin(\phi)$. Based on $z_{NaCl}$, we approached the tip close to the NaCl surface and obtained the data as a function of $r$.

Lateral manipulation is extremely sensitive to the crystallography and the tip apex, and the presence of tip could lower the energy barrier[33]. To estimate the effect of tip on the energy barrier, we derived the ratio of forces along the polar direction to along the non-polar

direction as a function of $r$ (Supplementary Fig. 4a), from the following equation using data in Fig. 4b.

$$\frac{F_{TH}^{P}}{F_{TH}^{NP}} = \frac{F_{T}\cos(\theta_{P})}{F_{T}\cos(\theta_{NP})} = \frac{r/\cos(\theta_{NP})}{r/\cos(\theta_{P})} = \frac{\Gamma_{NP}}{\Gamma_{P}} \tag{4}$$

This ratio includes the effect of tip on the forces, which increases as the tip approaches the molecule. In our system, threshold forces along the polar and non-polar directions affected by the tip can be described as $F_{TH}^{NP} = F_{TH}^{NP,0} + \Delta F$ and $F_{TH}^{P} = F_{TH}^{P,0} + k\Delta F$. A threshold frictional force ($F_{TH}$) is divided into an intrinsic threshold force ($F_{TH}^{0}$) and an external force ($\Delta F$, $k\Delta F$) exerted by the tip. Tip-molecule attraction reduces a normal force associated with a frictional force. Under the given condition of lateral manipulation, the reduced normal force is the same in both the polar and non-polar directions. $\Delta F$ is a tip-induced reduction in a frictional force along the non-polar direction. We assume that the reduced frictional force along the polar direction is proportional to that along the non-polar direction, $k\Delta F$. Using attractive part of the force derived from Morse potential[43], $F_{TH}^{NP} = F_{TH}^{NP,0}\exp(-x/\lambda_{NP})$ and $F_{TH}^{P} = F_{TH}^{P,0}\exp(-x/\lambda_{P})$, where $x$ is a molecule–substrate distance. As the tip reduces the normal force of the molecule, $x$ decreases with increasing a tip-molecule distance ($r$);

Since $z = r + x$, an analytic form of the ratio in terms of $r$ becomes

$$\frac{F_{TH}^{P}}{F_{TH}^{NP}} = \frac{F_{TH}^{P,0} + k\Delta F}{F_{TH}^{NP,0} + \Delta F} = \frac{F_{TH}^{P,0} + kF_{TH}^{NP,0}\{\exp(a\cdot r + b)-1\}}{F_{TH}^{NP,0}\exp(a\cdot r + b)} \tag{5}$$

$$= k + \left(\frac{F_{TH}^{P,0} - kF_{TH}^{NP,0}}{F_{TH}^{NP,0}}\right)\exp(-a\cdot r - b). \tag{6}$$

We obtained $k = 1.0462$, $a = 0.00505$ and $b = -1.1630$ by applying this equation to Supplementary Fig. 4a. As the tip approaches the water molecule, the threshold frictional forces along both the polar and the non-polar directions decrease along with the energy barriers (Supplementary Fig. 4b, c). $F_{TH}^{P}/F_{TH}^{NP}$ is also calculated to be 1.1564 using a sinusoidal model without the consideration of tip, which is close to the value of $k$. It should be noted that this simple model is not appropriate for describing the lateral manipulation at a longer distance than the experimental range (blue areas in Supplementary Fig. 4) because the tip effect on the molecule becomes negligible at a long tip-molecule distance. In our experimental results, single water molecules were easily dragged by the tip at a shorter tip-molecule distance. This clearly shows that the tip lowered the energy barriers of the lateral manipulation along the polar and non-polar directions.

### Computational details

We performed DFT calculations with the Perdew-Burke-Ernzerhof (PBE) exchange-correlation functional and the generalized gradient approximation (GGA) using the Vienna ab initio Simulation Package (VASP, 5.4.4 version)[44–46]. In addition, the D3 correction for the correction of the dispersive interactions was adopted[47,48]. The plane-wave cut-off was set to 400 eV with the convergence criterion for the self-consistent field iterations set to $10^{-5}$ eV. Vacuum slabs with more than 15 Å were used and the Brillouin zone was sampled with a k-points density of 0.03 Å$^{-1}$. We performed optimization of the lattice cell and atomic positions until the convergences of the forces on all atoms less than 0.01 eV Å$^{-1}$.

We built up two types of NaCl systems: the terrace and the step. The NaCl terrace consists of one $H_2O$ molecule on NaCl(100) with/without Ag(100) substrate, and the NaCl step includes one $H_2O$ molecule on NaCl(510) used for periodicity. As shown in Fig. 3c, f (the energy profile for translation of $H_2O$ on NaCl with/without the substrate), the calculated energy profiles of lateral manipulation of a water molecule on a 2 ML NaCl terrace with/without Ag substrate are nearly same, which indicates that the effect of Ag substrate on the adsorption

of water molecule is negligible. Thereof, we excluded the substrate when calculating the molecular adsorption at the NaCl step.

The adsorption energy ($E_{ads}$) of a water molecule on a NaCl surface was calculated by $E_{ads} = E_{Total} - E_{NaCl} - E_{H_2O}$, where $E_{Total}$, $E_{NaCl}$ and $E_{H_2O}$ are the energies of a $H_2O$ molecule on NaCl, bare NaCl substrate and a $H_2O$ molecule in vacuum, respectively.

The charge density difference of $H_2O$ adsorbed at $Na^+$ and $Cl^-$ ions of the step (state 3 and state 11 in Supplementary Fig. 5c) was calculated. The difference is defined by $\Delta\rho = \rho_{Total} - \rho_{NaCl} - \rho_{H_2O}$, where $\rho_{Total}$, $\rho_{NaCl}$ and $\rho_{H_2O}$ are the charge densities of $H_2O$ molecule on NaCl, bare NaCl substrate and a $H_2O$ molecule in vacuum, respectively. The charge densities were obtained by re-calculating $\rho_{NaCl}$ and $\rho_{H_2O}$ with the fixed atomic positions in $H_2O$ on NaCl system, respectively.

The simulated STM images were calculated using the DFT results. We made four models: the intact step, one $Na^+$ vacancy, one $Cl^-$ vacancy, and one NaCl vacancy of the step (Supplementary Fig. 8). After the structure relaxation, we used the Tersoff-Hamann model to simulate the STM images[49]. We obtained the images with the constant-current mode and the current is calculated by $n \approx 2 \times 10^{-4}\sqrt{I}$, where $n$ in Å is the charge density and $I$ in nA is the current[50,51].

### Reporting summary

Further information on research design is available in the Nature Portfolio Reporting Summary linked to this article.

### Data availability

The data supporting the findings of this study are available from the corresponding author upon request.

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

## Acknowledgements

This work was supported by the National Research Foundation of Korea (NRF), funded by the Ministry of Science, ICT & Future Planning (Grant No. NRF-2018R1A2B6006423 (H.-J.S.) and 2021R1A2C2006219 (H.-J.S.)) and by the Institute for Basic Science (IBS-R019-D1 (H.-J.S.)).

## Author contributions

H.-J.S. conceived and supervised the project. H.H. and Y.K. performed STM experiments. Y.P. and F.D. carried out the DFT calculations. H.H. and H.-J.S. wrote the manuscript. All authors revised and commented on the manuscript.

## Competing interests

The authors declare no competing interests.
