## [Peer Review File · Nature Communications]

Controlled dissolution of a single ion from a salt interfaceEditorial Note: Parts of this Peer Review File have been redacted as indicated to remove third-party material where no permission to publish could be obtained.

REVIEWER COMMENTS

Reviewer #1 (Remarks to the Author):

Han et al. Controlled dissolution of a single ion from salt, Nature Comm. Submitted

The authors report fascinating experiments in which STM tip is used to manipulate single water molecules on NaCl surfaces and step edges. The NaCl is prepared as two monolayers of ions, plus half of a third monolayer over Ag (100). The technique allows authors to probe the corrugated potential energy landscape felt by water molecules on step edges and terraces of the NaCl crystal. DFT calculations are used to help interpret the STM results, e.g. in terms of electrostatics, geometry, and solvent/ion polarization effects. The key finding is that, by pulling a water molecule along the step edge, the Cl⁻ ion is preferentially dislodged (as opposed to dislodging a Na⁺ ion). I have a few questions and suggestions for the authors, but otherwise the work warrants publication.

- 1) In Figure 4, what happens to the dislodged Cl⁻ ion? Why don't we see it anywhere in the images?
- 2) The authors need to discuss the important difference between a crystal surface in contact with an aqueous solution and the crystal surface with an adsorbed water molecule. The two situations are quite different, e.g. the single water molecule cannot solvate the dislodged ions, so "dissolution" might not be the most appropriate terminology (see point 3).
- 3) I think the real process being probed here is the 1D nucleation of kink sites along a step edge. The authors have shown that the removal of Cl⁻ to create two Na⁺ kinks is easier than the removal of Na⁺ to create two Cl⁻ kinks. Please cite and discuss the work of Christoffersen et al. J. Crystal Growth, (1991) in connection with these two processes.
- 4) Please discuss and distinguish the two main steps in dissolution. First an ion must be removed from a step edge to create a pair of kinks. Then the kinks recede from each other by alternating removal of cations and anions from the kink sites. Note that Joswiak et al. Cryst. Growth & Des. (2018) provide a kinetic model and calculations that include both steps.
- 5) The results in Figure 2 and Figure 3 will be quite intriguing to a large community of theorists who study non-equilibrium work theorems. The authors should call attention to the connection by citing Bustamante et al. Phys. Today (2005) and Jarzynsky, Ann. Rev. Cond. Matt. Phys. (2011).

Signed,
Baron Peters

Reviewer #2 (Remarks to the Author):

The authors present an impressive experimental STM and computational DFT study of water molecules on the surface of a thin NaCl film with steps. Manipulation of the water molecules along two different directions on the NaCl surface lattice leads to preferential motion across a row of Na ions or across a row with alternating Na and Cl ions. The authors provide STM and DFT evidence that water molecules moving across Cl ions interact strongly with these ions through their large polarizability. This moves charge density from the Cl ions towards the water and away from the neighboring Na ions, thereby breaking the NaCl bond and releasing Cl ions from the step.

The authors point out that water molecules prefer to interact with Na ions on the surface or on steps, primarily due to strong electrostatic interactions. It would have been a nice point of discussion if the

authors described how their results lead to a mechanism for dissolution of the anion in the absence of STM manipulation of water molecules which force them to interact with chloride ions. Presumably, this would involve the much higher density of water molecules interacting with a NaCl crystal surface under common dissolution conditions at STP. These crowded molecular conditions would lead some water molecules to naturally interact with chloride ions, which could trigger the dissolution mechanism described by the authors for single water molecules at 20 K in UHV conditions. In this context, the authors could address the following questions. How do the energies calculated from DFT for the various possible configurations (and the resulting differences thereof) compare to the energies that set the scale for water's characteristics? That is, hydrogen bond enthalpy, $k_B T$, etc. Does removing a Cl ion from the step and having it readsorb nearby result in a change in entropy? What exactly were the fates of all the Cl ions that were dissolved in this experiment; did they fly off into the vacuum or did they readsorb?

The clarity of the manuscript could use some work. What the data is, why it was gathered, and why it is analyzed the way it is are not always apparent. For instance, figure 2 is information dense but could be improved. Let me assume that the lower plots in fig 2c are meant to be the best description of the monolayer topology possible using STM. They are the paths or the 'landscape' that the water molecule will be dragged along, correct?

In the blue trace the wave crests are Cl and the troughs are Na. In the red trace I will assume the crests are Na and the troughs are the Cl-Cl midpoints. If this is the case then the figure could be edited to include some of this information, either by use of shaded areas to indicate atom type or other graphical notation. This sets the stage for the reader, and in terms of narrative ordering, presenting the 'lay of the land' first is preferable. Then a water molecule is dragged along and the effect of that landscape on the tip-molecule complex is measured. This is the noisier asymmetric trace. Shaded areas that show atom speciation should be included here to highlight the cause and effect relationship.

In 2c what are 'surface corrugations', the height of the sample in constant tunneling current without a water molecule attached to the tip? If the 'surface corrugation' is not taken on exactly the path the water molecule is dragged over, then why not? And if so, how are these traces aligned along the x-axis in the figure? If they are not aligned, could they be, such that the cause-and-effect relationships are clear?

Extended data Figures 3 and 7 illustrate the likely configuration of the water molecule as it is dragged. These DFT illustrations should be included in the main text by integrating with appropriate main text figures. The editor should encourage the inclusion of another figure (five main text figures) and the reorganization of the content of these figures.

There are a few minor points that would help to clarify the authors' presentation, but otherwise this is a very interesting paper that will likely interest the readers of Nature Communications.

Authors should state the sequence of atoms that the tip passes over when moving along the polar and non-polar directions.

When DFT results are shown in a figure, it should be stated in the caption that it is a DFT result.

Authors should state what the white dots in Fig. 2d represent.

It would be useful to explain the shape of the curve in Fig. 3a and why the angle ϕ is measured at the jump in tip height.

The text states "The angle clearly increased with decreasing r , implying that U increased (Supplementary Fig. 1)." However, within error bars I do not see any changes in Supplementary Fig. 1. Does a Welch's t-test suggest the trend is significant?

The point about higher energy barrier but lower friction (lines 131 and 132) needs some explanation in the main text.

How is the readsorbed chloride ion evident in Fig. 4b?

Orange is used to represent O in Fig. 4e/g, not red.

The latter part of the following sentence, starting with "becomes redistributed", is unclear to me. "The

electron cloud of the anion is concentrated towards the cation, and hence the charge of the anion by the polarizable interaction becomes redistributed against the electron transfer from Na atom to Cl atom.”

The ‘glitched’ water molecule in fig1b should be explained in one or two sentences.

Line 79 says ‘(OH stretching mode of water molecule)’ but I did not understand why this was said.

Line 80 says ‘reduced tip-molecule distance’ but it is not clear what it is reduced from (from the typical imaging distances I might assume).

Figure 1: Include same-size scale bar for all three images (a,b,c). Suggested size is 4-5 nm or some integer multiple of the chlorine lattice spacing. It would be nice if Fig 1d could indicate the Na-Cl distance.

Fig1b uses artificial color to emphasize water molecules adsorbed onto sites where chlorines are missing from the lattice (caption does not describe white lines, but main text does). How is it apparent to the authors that chlorines are absent from these locations?

Text and captions could remind the reader that the bright spots in the images are chlorine atoms, whereas the ‘shadows’ are sodium. Likewise, more scale bars in STM-images and DFT-generated images would also allow the reader to come to that conclusion organically.

Reviewer #3 (Remarks to the Author):

This manuscript presents interesting results of water molecule interactions on NaCl surface. The main observation is that the Cl⁻ ion will dissolve more easily than Na⁺ ion while moving a single water molecule next to an atomic step on the surface. The water molecule is moved with an STM tip. Also, different interaction energies are observed when dragging a water on the polar or non-polar direction. The authors explain the difference of dissolution of Cl and Na with the polarization of these two ions.

The water-NaCl interaction is an intensively studied topic but there are not too many atomic-level experimental studies. This manuscript is among the few studies of this type and some other are included to the ref of this MS. I find the approach that a single water molecule is dragged along the surface a quite artificial setup. The results are interesting but how much they will tell from the liquid water NaCl interaction. I see that this manuscript can be published in a good Physical Chemistry journal, like J. Phys. Chem. C, PCCP or J. Chem. Phys. but I do not see that the results are sufficiently important for Nature Comm.

Reviewer #4 (Remarks to the Author):

Reply to Reviewers' Comments

We are grateful for the reviewers' valuable and constructive comments on the manuscript. Specific responses to the reviewer are as follows.

Reviewer #1 (Remarks to the Author):

Han et al. Controlled dissolution of a single ion from salt, Nature Comm. Submitted

The authors report fascinating experiments in which STM tip is used to manipulate single water molecules on NaCl surfaces and step edges. The NaCl is prepared as two monolayers of ions, plus half of a third monolayer over Ag(100). The technique allows authors to probe the corrugated potential energy landscape felt by water molecules on step edges and terraces of the NaCl crystal. DFT calculations are used to help interpret the STM results, e.g. in terms of electrostatics, geometry, and solvent/ion polarization effects. The key finding is that, by pulling a water molecule along the step edge, the Cl^- ion is preferentially dislodged (as opposed to dislodging an Na^+ ion). I have a few questions and suggestions for the authors, but otherwise the work warrants publication.

We thank Reviewer 1 for the appreciation of our work and very positive statements about it.

[Comment 1]

In Figure 4, what happens to the dislodged Cl^- ion? Why don't we see it anywhere in the images?

The protrusion in Fig. 4c (revised Fig. 5c) is a dissolved Cl^- ion on the upper terrace. However, the dislodged Cl^- ion was not present on the surface or STM tip in Fig. 4d (revised Fig. 5d). We believe that the extracted Cl^- ion desorbed and escaped into vacuum.

[Comment 2]

The authors need to discuss the important difference between a crystal surface in contact with an aqueous solution and the crystal surface with an adsorbed water molecule. The two situations are quite different, e.g. the single water molecule cannot solvate the dislodged ions, so "dissolution" might not be the most appropriate terminology (see point 3).

The primary differences between “salt in liquid water” and “a water molecule on the salt” are temperature and composition. Composition is related to the entropy of the system, which competes with the change in enthalpy. When the salt comes in contact with an aqueous solution, numerous water molecules dissolve the salt crystal, driven by the entropy of the system. Eventually, the ionic bond weakens and is broken by electrostatic interactions with water molecules, which is difficult to evaluate experimentally.

In this study, this problem was simplified by placing isolated water molecules on the surface of NaCl as a model system. As demonstrated in the manuscript, we found that the strong anion–water interaction at a step induces charge depletion between the ionic bonds, leading to preferential dissolution of the anion. Our results elucidated the very early stage of the dissolution process from a microscopic perspective. Regarding appropriate terminology, we believe that ‘dissolution’ is the best description for this process as it is triggered by the interaction between the dipole moment of a polar molecule and ions.

[Comment 3]

I think the real process being probed here is the 1D nucleation of kink sites along a step edge. The authors have shown that the removal of Cl^- to create two Na^+ kinks is easier than the removal of Na^+ to create two Cl^- kinks. Please cite and discuss the work of Christoffersen et al. *J. Crystal Growth*, (1991) in connection with these two processes.

We appreciate the suggestions of Reviewer 1. Christoffersen et al. [*J. Cryst. Growth* **113**, 599-605 (1991)] proposed that the interfacial surface tension of a solid electrolyte (i.e., salt) is proportional to the negative logarithm of its solubility, which is derived based on surface nucleation under supersaturation conditions. They did not need to consider unknown parameters, such as entropy and hydration number. They predicted that the charge type of the surface ions would contribute differently to the surface tension.

Despite the favourable adsorption of water molecule at Na^+ ion, our results revealed the preferential dissolution of a single Cl^- ion over a single Na^+ ion. The formation energies of vacancy at Na^+ and Cl^- sites without water molecules are 4.86 and 5.01 eV, respectively [*Phys. Rev. B* **52**, 11424-11431 (1995)]. The extraction of Na^+ ions, which creates two Cl^- kinks, is energetically favourable. However, when water molecules are involved in the dissolution

process, the strong polarization interactions between the anions and water molecules weaken the ionic bonds and facilitate the preferential dissolution. We discussed the microscopic aspect of the dissolution process and the references, as Reviewer 1 commented, in the revised manuscript.

[Comment 4]

Please discuss and distinguish the two main steps in dissolution. First an ion must be removed from a step edge to create a pair of kinks. Then the kinks recede from each other by alternating removal of cations and anions from the kink sites. Note that Joswiak et al. *Cryst. Growth & Des.* (2018) provide a kinetic model and calculations that include both steps.

Joswiak et al. [*Cryst. Growth Des.* **18**, 723-727 (2018)] established the nonequilibrium model of step velocity in relation to the growth rate of a crystal. They derived the supersaturation dependence of the kinetic kink density by applying a one-dimensional nucleation approach. The key parameters for the kink nucleation and annihilation rates were low kink energy and high supersaturation, implying that the steady-state movement of multi-height kinks closely approximated the actual propagation of a step. We believe that our single-molecule study represents a very early stage of the dissolution process, that is, the nucleation of kink sites. We discussed the relationship of the microscopic picture of our results to the macroscopic dissolution process in terms of the nucleation and propagation of kinks in the revised manuscript.

[Comment 5]

The results in Figure 2 and Figure 3 will be quite intriguing to a large community of theorists who study non-equilibrium work theorems. The authors should call attention to the connection by citing Bustamante et al. *Phys. Today* (2005) and Jarzynsky, *Ann. Rev. Cond. Matt. Phys.* (2011).

We agree with Reviewer 1 that our findings could be intriguing to researchers studying nonequilibrium work theorems. We added references to the revised manuscript according to Reviewer 1's suggestions.

Reviewer #2 (Remarks to the Author):

The authors present an impressive experimental STM and computational DFT study of water molecules on the surface of a thin NaCl film with steps. Manipulation of the water molecules along two different directions on the NaCl surface lattice leads to preferential motion across a row of Na ions or across a row with alternating Na and Cl ions. The authors provide STM and DFT evidence that water molecules moving across Cl ions interact strongly with these ions through their large polarizability. This moves charge density from the Cl ions towards the water and away from the neighboring Na ions, thereby breaking the NaCl bond and releasing Cl ions from the step.

We thank Reviewer 2 for considering our results impressive and for providing helpful comments.

[Comment 1]

The authors point out that water molecules prefer to interact with Na ions on the surface or on steps, primarily due to strong electrostatic interactions. It would have been a nice point of discussion if the authors described how their results lead to a mechanism for dissolution of the anion in the absence of STM manipulation of water molecules which force them to interact with chloride ions. Presumably, this would involve the much higher density of water molecules interacting with a NaCl crystal surface under common dissolution conditions at STP. These crowded molecular conditions would lead some water molecules to naturally interact with chloride ions, which could trigger the dissolution mechanism described by the authors for single water molecules at 20 K in UHV conditions. In this context, the authors could address the following questions. How do the energies calculated from DFT for the various possible configurations (and the resulting differences thereof) compared to the energies that set the scale for water's characteristics? That is, hydrogen bond enthalpy, $k_B T$, etc.

The primary differences between the “salt in liquid water” at standard temperature and pressure and the “water molecule on the salt” of our system are temperature and composition. Composition is related to the entropy of the system, which competes with the change in enthalpy. When the salt comes in contact with an aqueous solution, numerous water molecules dissolve the salt crystal, driven by the entropy of the system. Eventually, the ionic bonding

weakens and is broken by the electrostatic interaction with the water molecules, which is difficult to evaluate experimentally.

In this study, this problem was simplified by placing isolated water molecules on the surface of NaCl as a model system. As demonstrated in the manuscript, we found that the strong anion–water interaction at a step drives charge depletion between ionic bonds, resulting in preferential dissolution of the anion. Our results elucidated the very early stage of the dissolution process from a microscopic perspective. We also discussed the relationship of this microscopic picture of our results to the macroscopic dissolution process in the revised manuscript.

In response to the additional questions raised by Reviewer 2, it should be noted that extensive theoretical studies have been conducted on the energies of potential hydrated configurations. The strength of hydrogen bonding between water molecules varies significantly, ranging from 0.3 to 11.6 kcal/mol in water clusters [J. Phys. Chem. A **124**, 6699-6706 (2020)]. This wide range is attributed to cooperativity between water molecules, which can either reinforce or weaken hydrogen bonds. The water dimer, which is a model cluster linked by a single hydrogen bond, is unaffected by this cooperative interaction. The formation enthalpy of the water dimer was calculated to be -3.17 kcal/mol at 375 K [J. Phys. Chem. **100**, 2993-2997 (1996)]. In the case of an anion–water cluster, a hydrogen bond is formed between the partially positive H atom and the negative anion [J. Phys. Chem. **100**, 9703-9713 (1996) and J. Chem. Phys. **113**, 5259-5272 (2000)]. For a $\text{Cl}^-(\text{H}_2\text{O})_1$ cluster, one OH bond points towards the Cl^- ion, while the other bond remains free. The formation enthalpies of these hydrogen bonds were calculated to be -14.4 and -16.07 kcal/mol, respectively [J. Phys. Chem. **100**, 9703-9713 (1996) and J. Chem. Phys. **113**, 5259-5272 (2000)]. For $\text{Cl}^-(\text{H}_2\text{O})_n$ clusters with $n = 2, 3,$ and 4 , the formation enthalpy of hydrogen bonds changes with the number of water molecules: $-30.85,$ $-45.05,$ and -57.53 kcal/mol, respectively. The strength of the hydrogen bonds per water molecule is significantly higher in the hydrated Cl^- ions than in the water clusters.

[Comment 2]

Does removing a Cl ion from the step and having it readsorb nearby result in a change in entropy? What exactly were the fates of all the Cl ions that were dissolved in this experiment; did they fly off into the vacuum or did they readsorb?

The process of extracting a Cl^- ion and its subsequent adsorption on the surface increases the entropy of the system. Considering the elastic deformation caused by the vacancy pair, the entropy of formation of a Schottky defect in an ionic crystal was calculated to be $5.86k_B$ [Phys. Rev. **144**, 738 (1966)] and $10k_B$ [J. Physique. Lett. **36**, 9-12 (1975)]. The creation of a vacancy and an adatom at the surface, as well as in the bulk, result in an increase in entropy.

Both readsorption and desorption of the dissolved Cl^- ion were observed in the revised Figs. 5c and 5d, respectively. In the latter case, the dissolved Cl^- ion was not present on the surface or STM tip. We believe that the extracted Cl^- ion desorbed and escaped into vacuum.

[Comment 3]

The clarity of the manuscript could use some work. What the data is, why it was gathered, and why it is analyzed the way it is are not always apparent. For instance, figure 2 is information dense but could be improved. Let me assume that the lower plots in fig 2c are meant to be the best description of the monolayer topology possible using STM. They are the paths or the ‘landscape’ that the water molecule will be dragged along, correct?

In the blue trace the wave crests are Cl and the troughs are Na. In the red trace I will assume the crests are Na and the troughs are the Cl-Cl midpoints. If this is the case then the figure could be edited to include some of this information, either by use of shaded areas to indicate atom type or other graphical notation. This sets the stage for the reader, and in terms of narrative ordering, presenting the ‘lay of the land’ first is preferable. Then a water molecule is dragged along and the effect of that landscape on the tip-molecule complex is measured. This is the noisier asymmetric trace. Shaded areas that show atom speciation should be included here to highlight the cause and effect relationship.

In 2c what are ‘surface corrugations’, the height of the sample in constant tunneling current without a water molecule attached to the tip? If the ‘surface corrugation’ is not taken on exactly the path the water molecule is dragged over, then why not? And if so, how are these traces aligned along the x-axis in the figure? If they are not aligned, could they be, such that the cause-and-effect relationships are clear?

Extended data Figures 3 and 7 illustrate the likely configuration of the water molecule as it is dragged. These DFT illustrations should be included in the main text by integrating with appropriate main text figures. The editor should encourage the inclusion of another figure (five

main text figures) and the reorganization of the content of these figures.

There are a few minor points that would help to clarify the authors' presentation, but otherwise this is a very interesting paper that will likely interest the readers of Nature Communications. Authors should state the sequence of atoms that the tip passes over when moving along the polar and non-polar directions. When DFT results are shown in a figure, it should be stated in the caption that it is a DFT result.

Thank you for Reviewer 2's comments regarding the clarity of the figures. The lower plots in Fig. 2c in the original manuscript represent the 'surface corrugation' of the clean NaCl(100) surface along the lines marked in Fig. 2d. Surface corrugation refers to the height profile of the surface in the constant tunnelling current mode without a water molecule, as Reviewer 2 noted. Although the surface corrugation was not taken on exactly the path over which the water molecule was dragged, it did not matter because the atomic corrugation did not change place-to-place on the same surface. It should also be noted that the x-axis of Fig. 2c is not the x-coordinate but the displacement; thus, alignment is not required. As Reviewer 2 noted, only Cl^- ions were imaged as protrusions in the STM images owing to its high electron density of states near the Fermi level.

Based on the suggestions of Reviewer 2, we improved the clarity of the manuscript. The Na^+ and Cl^- ions were superimposed on the STM image (Fig. 2d). Extended data Fig. 3 has been included in the main text. The results of the DFT calculations were also stated in the captions.

[Comment 4]

Authors should state what the white dots in Fig. 2d represent.

The white lattice in Fig. 2d represents the Cl^- ion. We explained the Fig. 2d in the revised manuscript.

[Comment 5]

It would be useful to explain the shape of the curve in Fig. 3a and why the angle phi is measured at the jump in tip height.

The curve in Fig. 3a (revised Fig. 4a) represents the trace of the tip during the lateral manipulation. The periodic array of atoms of the substrate yielded an equal displacement for each movement in the tip-height trace. A sudden jump in the trace implies that the water molecule is attracted by the tip and hops by one adsorption site towards the tip. Subsequently, the tip moves over the top and then downwards along the contour of the water molecule. Once the tip moves to the next-nearest adsorption site, attractive interaction again pulls the molecule below the tip. This shape of tip-height trace is characteristic of the pulling of a molecule [Phys. Rev. Lett. **79**, 697-700 (1997)].

Lateral motion is initiated when the lateral component (F_L) of the tip-molecule attractive force (F_T) exceeds the threshold frictional force between the particle and surface (F_{TH}) [Science **319**, 1066-1069 (2008)]. At the moment of jump, the lateral force reaches the threshold frictional force, and the geometry between the tip and particle in the revised Fig. 4a offers the relation $F_L = F_T \times \cos(\phi) = F_{TH}$, with an angle (ϕ) between F_T and F_L . This geometry is reflected in the tip height traces, enabling an estimation of the threshold frictional forces from the angles [Appl. Phys. Lett. **99**, 221902 (2011)].

[Comment 6]

The text states “The angle clearly increased with decreasing r , implying that U increased (Supplementary Fig. 1).” However, within error bars I do not see any changes in Supplementary Fig. 1. Does a Welch’s t-test suggest the trend is significant?

Since the angle is correlated with the tip-molecule distance, we prepared Supplementary Fig. 1 in the previous manuscript. As Reviewer 2 mentioned, however, it is not easy to find a change in angles between ‘230’ and ‘300’ pA in Supplementary Fig. 1b in the original manuscript. Welch’s t-test for these currents at a significance level of 0.05 yields a large p-value of 0.68873 and a larger critical two-tail critical value (2.04523) than the test statistic (-0.40461), indicating that there is no statistically significant difference. To ensure the correlation between the angle and the tip-molecule distance, we added additional data obtained at 350 and 420 pA to the revised manuscript, which clearly shows the correlation between the tip-molecule distance and angle (revised Supplementary Fig. 3).

[Comment 7]

The point about higher energy barrier but lower friction (lines 131 and 132) needs some explanation in the main text.

We have explained the results of a higher energy barrier but lower frictional force in the original manuscript (lines 133–138) and the original Supplementary Information (pages 5 and 6). Fig. 3c (revised Fig. 4c) illustrates that a higher energy is required to move the molecule along the non-polar direction, whereas Fig. 3b (revised Fig. 4b) indicates a lower threshold frictional force along this direction. The energy barriers for surface hopping (E_b) were calculated to be 164 and 201 meV along the polar and non-polar directions, respectively (revised Fig. 3). To explain the experimental results, we employed a simple sinusoidal model for the surface potential. The surface potential along the non-polar or polar directions can be described as a periodic potential: $V(x) = \frac{1}{2} E_b \cos(2\pi x/a_0)$, where a_0 is the spacing between the nearest Na^+ sites. The threshold frictional forces ($F_{\text{TH}} = (dV/dx)|_{\text{max}}$) were calculated to be 207 and 179 pN along the polar and non-polar directions, respectively. This simple model clearly explains the higher energy barrier and lower frictional force in the non-polar direction.

[Comment 8]

How is the readsorbed chloride ion evident in Fig. 4b?

It is not easy to resolve the re-adsorbed chloride ion in Fig. 4b (revised Fig. 5b) alone; however, its high-resolution image {Fig. 4c (revised Fig. 5c)} clearly shows a vacant Cl^- site at the step. Since there was no bright feature before the manipulation, we conclude that this feature is a readsorbed chloride ion.

[Comment 9]

Orange is used to represent O in Fig. 4e/g, not red.

Although we assigned the red colour to oxygen, some oxygen atoms appeared orange in Fig. 4e and g (revised Fig. 5e and g), as Reviewer 2 noted. This is due to the overlap of the red and yellow colours which represent the accumulation of electrons.

[Comment 10]

The latter part of the following sentence, starting with “becomes redistributed”, is unclear to me. “The electron cloud of the anion is concentrated towards the cation, and hence the charge of the anion by the polarizable interaction becomes redistributed against the electron transfer from Na atom to Cl atom.”

An ionic bond is formed by the electron transfer from a Na atom to a Cl atom. In addition, the electron cloud of the anion is concentrated towards the cation in the ionic compound because of the higher polarizability of the anion [J. Chem. Phys. **111**, 2308-2049 (1999)]. Therefore, we would like to say that the direction of charge transfer is opposite to that of polarization.

[Comment 11]

The ‘glitched’ water molecule in fig1b should be explained in one or two sentences.

The glitched water molecule in Fig. 1b represents the hopping of a single water molecule during imaging, which is not relevant to the main topic. We replaced Fig. 1b with a new figure that clearly shows the adsorption site for a water molecule on the NaCl surface in the revised manuscript.

[Comment 12]

Line 79 says ‘(OH stretching mode of water molecule)’ but I did not understand why this was said.

When tunnelling electrons flow through a molecule below the tip, the inelastic electron-tunnelling process induces vibrational excitation of the molecule, which can result in hopping, desorption, and other reactions [Science **295**, 20055-2058 (2002)]. While manipulating a water molecule on the NaCl surface, the excitation of two vibrational modes (bending (198 meV) or stretching mode (445 meV)) can trigger the hopping of the molecule [Nat. Mat. **9**, 442-447 (2010)]. The hopping probability at a tunnelling current of 1.4 nA by the excitation of the bending mode was extremely low (10^{-10}), whereas the hopping probability by the excitation of the stretching mode increased to close to 10^{-7} (Fig. R1). We focused on avoiding any vibrationally excited motion of water molecule during lateral manipulation because the hopping direction is not easy to control. In our results, a sample bias of less than

450 mV at low tunnelling currents did not show any unwanted inelastic processes. Therefore, we stated “The molecules were laterally dragged with an STM tip at a sample bias lower than 450 mV (OH stretching mode of the water molecule) to avoid undesirable hopping or desorption” to clarify that the molecule was not vibrationally excited during the lateral manipulation.

Fig. R1 | Hopping reaction of a single water molecule. **a**, STM images of water molecules on 2-ML-thick NaCl ($10 \times 10 \text{ nm}^2$, $V_s = -200 \text{ mV}$ and $I_t = 50 \text{ pA}$), before (upper) and after (lower) applying bias voltages at the marked water molecules. **b**, Action spectrum for the hopping reaction of a water molecule at the tunnelling current of 1.4 nA. The solid line is a fit to the data. **c**, Hopping rates as a function of the tunnelling rate current for -400 and -480 mV . The solid lines are least-squares fits to the data and correspond to the power laws.

[Comment 13]

Line 80 says ‘reduced tip-molecule distance’ but it is not clear what it is reduced from (from the typical imaging distances I might assume).

The tip-molecule distance was reduced from the typical imaging distance, as Reviewer 2 noted. We modified this expression in the revised manuscript.

[Comment 14]

Figure 1: Include same-size scale bar for all three images (a,b,c). Suggested size is 4-5 nm or some integer multiple of the chlorine lattice spacing. It would be nice if Fig 1d could indicate the Na-Cl distance.

In the revised manuscript, we included the scale bars in Fig. 1 and the lattice constant of the NaCl crystal in Fig. 1e, as suggested by Reviewer 2.

[Comment 15]

Fig. 1b uses artificial color to emphasize water molecules adsorbed onto sites where chlorines are missing from the lattice (caption does not describe white lines, but main text does). How is it apparent to the authors that chlorines are absent from these locations? Text and captions could remind the reader that the bright spots in the images are chlorine atoms, whereas the ‘shadows’ are sodium. Likewise, more scale bars in STM-images and DFT-generated images would also allow the reader to come to that conclusion organically.

We appreciate the Reviewer 2’s comments. We explained that chlorine ions were imaged as protrusions in the STM image and added scale bars to Fig. 1 in the revised manuscript.

Reviewer #3 (Remarks to the Author):

This manuscript presents interesting results of water molecule interactions on NaCl surface. The main observation is that the Cl^- ion will dissolve more easily than Na^+ ion while moving a single water molecule next to an atomic step on the surface. The water molecule is moved with an STM tip. Also, different interaction energies are observed when dragging a water on the polar or non-polar direction. The authors explain the difference of dissolution of Cl and Na with the polarization of these two ions.

We thank the Reviewer 3 for considering our results interesting and for providing constructive comments on the work.

[Comment 1]

The water-NaCl interaction is an intensively studied topic but there are not too many atomic-level experimental studies. This manuscript is among the few studies of this type and some other are included to the ref of this MS. I find the approach that a single water molecule is dragged along the surface a quite artificial setup. The results are interesting but how much they will tell from the liquid water-NaCl interaction. I see that this manuscript can be published in a good Physical Chemistry journal, like J. Phys. Chem. C, PCCP or J. Chem. Phys. but I do not see that the results are sufficiently important for Nature Comm.

We would like to highlight the significance of our results for a broader scientific audience. A key challenge in chemistry and materials science is the manipulation of individual bonds in matter. Extensive research has been conducted on the visualisation and manipulation of covalent bonds [Phys. Rev. Lett. **78**, 4410 (1997), Science **286**, 1719 (1999), Nat. Mater. **9**, 442 (2010), Science **360**, 521 (2018)] and metallic bonds [Sci. Adv. **6**, eaay5849 (2020) and Science **371**, 498 (2021)]. However, the cleavage of ionic bonds in salts by water molecules has never been visualised or manipulated, which we have achieved in this study. Despite the importance of salt dissolution as a fundamental process, its precise mechanism and dynamics remain controversial. One of the reasons for the ambiguity of the salt dissolution process is the lack of an atomic picture. Because dissolution occurs in solution, it has been difficult to probe single ion or salt dissolution. In this study, we circumvented this problem using ultrathin NaCl films and single water molecules in an ultrahigh vacuum (UHV) at a low temperature (4.4 K),

which enabled us to investigate the dynamics of single water molecules on the NaCl surface by means of LT-STM. We also discussed the relationship of this microscopic picture of our results to the macroscopic dissolution process in the revised manuscript.

We hope that Reviewer 3 finds our results to be of significant general interest to the diverse readership of Nature Communications.

List of Change

* In the revised manuscript

Most of the extended data figures in the original manuscript have been moved to the supplementary information according to formatting instructions of Nature Communications. Extended data Figs. 1 and 3 have been incorporated into the main figures in the revised manuscript, according to referee's comments.

Additionally, the physical height of the 3-ML-thick NaCl film, stated as '864' pm (Line 322) should be corrected to '846' pm. Accordingly, we have changed the tip-molecule distance (r) in Fig. 4a,b and Supplementary Fig.4a-c in the revised manuscript. However, this correction does not alter the conclusion that the non-polar direction has a higher energy barrier for lateral manipulation. The authors apologize for the oversight and any inconvenience it may have caused.

[1] page 2:

- by both temperature and composition^{9,10}.

[2] page 3:

- Only Cl⁻ ions were imaged as protrusions in STM images due to its high density of state near the Fermi level. In Fig. 1b, a single water molecule on the terrace was located at the hollow sites of Cl⁻ lattice (white dash)²⁶.
- a Na⁺ site (Fig. 1d)
- on the terrace and at the step, respectively (Fig. 1c,e).

[3] page 4:

- At a closer tip-molecule distance,
- the magnitude of bias voltage (Supplementary Fig. 1)²⁹.
- in Fig. 2c,d, which means that the polar direction is smoother than the non-polar direction.

[4] page 5:

- yielded the comparable result (Supplementary Fig. 1a).

- that aligns the dipole of the molecule towards the anion (Fig. 3b)
- obtained from the slope of the tip trace (Fig. 4a).
- the higher angle than along the polar direction (Fig. 2 and Supplementary Fig. 1b)

[5] page 6:

- the molecule on the NaCl surface (see Supplementary Fig. 2 and Methods).
- Figure 4b shows the changes in the angle as a function of tip–molecule distance.
- implying that U increased (Supplementary Fig. 3).
- Inserting the fitted lines in Fig. 4b into Eq. (2), we found a negative power-law relation between Γ and r , $\Gamma \propto r^{-n}$ with $n = 7.75$ and 6.61 in the non-polar and polar directions, respectively (Fig. 4c).
- a higher energy barrier along the non-polar direction (Fig. 3),
- (see Supplementary Fig. 4 and Supplementary text).
- calculated using the DFT results (Fig. 3) shows

[6] page 7:

- When manipulating a water molecule along the step (Supplementary Fig. 5),
- In Fig. 5a, two water molecules
- was also created on the step (Fig. 5b).
- dissociation during the manipulations (Supplementary Fig. 6).
- was re-adsorbed near the vacancy (Fig. 5b).
- Figure 5c shows a single Cl^- vacancy at the step.
- Figure 5d shows another NaCl step from which a Cl^- ion was extracted by manipulation (Supplementary Fig. 7).
- the extraction of a single Cl^- ion (Supplementary Fig. 8).

[7] page 8:

- Figure 5e–h show the charge density difference plots and its cross-sectional plots of a water molecule at the interfacial step.

- The water molecule adsorbed near Na⁺ ion (Fig. 5e,f and the state 3 in Supplementary Fig. 5c)
- At the interfacial Cl⁻ site (Fig. 5g,h and the state 11 in Supplementary Fig. 5c)

[8] page 9:

- This microscopic dissolution process demonstrated by single-molecule manipulation reflects the real dissolution process under ambient conditions, where numerous water molecules dissolve the salt spontaneously^{39,40}. In such crowded conditions, the adsorption of multiple water molecules can severely distort the electron cloud of a Cl⁻ ion compared to a single water molecule, thereby facilitating easier dissolution and hydration of the Cl⁻ ion. Once a Na⁺ kink is created, which offers additional adsorption sites with lower coordination number, ionic bonds become weaker, promoting the dissolution of ions^{41,42}. Eventually, the sequential dissolution proceeds as kinks propagate until the solubility limit is reached.

[9] pages 12, 13

39. Bustamante, C., Liphardt, J. & Ritort, F. The nonequilibrium thermodynamics of small systems. *Phys. Today* **58**, 43–48 (2005).

40. Jarzynski, C. Equalities and inequalities: irreversibility and the second law of thermodynamics at the nanoscale. *Ann. Rev. Condens. Matter Phys.* **2**, 329–351 (2011).

41. Christoffersen, J., Rostrup, E. & Christoffersen, M. R. Relation between interfacial surface tension of electrolyte crystals in aqueous suspension and their solubility; a simple derivation based on surface nucleation. *J. Cryst. Growth* **113**, 599-605 (1991).

42. Joswiak, M. N., Peters, B. & Doherty, M. F. Nonequilibrium kink density from one-dimensional nucleation for step velocity predictions. *Cryst. Growth Des.* **18**, 723-727 (2018).

43. Perdew, J. P., Burke, K. & Ernzerhof, M. Generalized gradient approximation made simple. *Phys. Rev. Lett.* **77**, 3865-3868 (1996).

44. Kresse, G. & Furthmüller, J. Efficient iterative schemes for *ab initio* total-energy calculations using a plane-wave basis set. *Phys. Rev. B* **54**, 11169-11186 (1996).

45. Kresse, G. & Furthmüller, J. Efficiency of *ab initio* total-energy calculations for metals

and semiconductors using a plane-wave basis set. *Comput. Mater. Sci.* **6**, 15-50 (1996).

46. Grimme, S., Antony, J., Ehrlich, S. & Krieg, H. A consistent and accurate *ab initio* parametrization of density functional dispersion correction (DFT-D) for the 94 elements H–Pu. *J. Chem. Phys.* **132**, 154104 (2010).
47. Grimme, S., Ehrlich, S. & Goerigk, L. Effect of the damping function in dispersion corrected density functional theory. *J. Comput. Chem.* **32**, 1456–1465 (2011).
48. Tersoff, J. & Hamann, D. R. Theory of the scanning tunneling microscopy. *Phys. Rev. B* **31**, 805–813 (1985).
49. Hofer, W.A., Redinger, J. & Varga, P. Modeling STM tips by single adsorbed atoms on W(100) films: 5d transition metal atoms. *Solid State Commun.* **113**, 245-250 (1999).
50. Hofer, W. A., Foster, A. S. & Shluger, A. L. Theories of scanning probe microscopes at the atomic scale. *Rev. Mod. Phys.* **75**, 1287–1331 (2003).

[10] page 14:

- by measuring *I*-*z* spectroscopy (Supplementary Fig. 2a).
- with the physical dimension (d_{NaCl} , 846 pm) of the 3 ML NaCl, we obtained the tip height on the film (z_{NaCl} , 390 pm) as follow;
- using a simple geometric relation in Fig. 4a

[11] page 15:

- Simulation Package (VASP, 5.4.4 version)⁴³⁻⁴⁵.
- the dispersive interactions was adopted^{46,47}.
- As shown in Fig. 3c and d
- The charge density difference of H₂O adsorbed at Na⁺ and Cl⁻ ions of the step (state 3 and state 11 in Supplementary Fig. 5c) was calculated.

[12] page 16:

- one NaCl vacancy of the step (Supplementary Fig. 8).
- Tersoff-Hamann model to simulate the STM images⁴⁸.

- I [nA] is the current^{49,50}.

[13] page 18:

- **Fig. 1 | Water molecules on a NaCl surface.** **a**, STM image of water molecules on 2 and 3 ML NaCl surfaces ($V_s = 200$ mV; sample bias and $I_t = 50$ pA; tunnelling current). **b**, High-resolution STM image of a single water molecule on the 2 ML NaCl surface detected with a water-terminated STM tip ($V_s = -200$ mV, $I_t = 50$ pA). Dotted lines represent Cl^- lattice. **c**, Top and side views of the optimized configuration of water molecule on the terrace calculated by DFT. **d**, Flattened high-resolution STM image of a single water molecule at the step ($V_s = 300$ mV, $I_t = 500$ pA). The inset shows an unprocessed image. **e**, Top and side views of the optimized configuration of water molecule at the step calculated by DFT. In **c** and **e**, blue, blue-green, red, and white spheres represent Na^+ ions, Cl^- ions, oxygen atoms and hydrogen atoms, respectively. Scale bars in **a**, **b** and **d** are 6, 1, and 1 nm, respectively.

[14] page 19:

- **Fig. 2 | Lateral manipulation of a single water molecule on the terrace.** **a,b**, STM images of a single water molecule on the 3 ML terrace before (left) and after a lateral manipulation (right) along the non-polar direction (blue arrow) and the polar direction (red arrow), respectively ($10 \times 5 \text{ nm}^2$, $V_s = 200 \text{ mV}$, $I_t = 50 \text{ pA}$). The molecules were manipulated under $V_s = 200 \text{ mV}$ and $I_t = 800 \text{ pA}$ at a speed of 50 pm/s . **c**, Tip-height traces of the molecules on the terraces (blue dots; non-polar direction, red dots; polar direction) and surface corrugations (blue line; non-polar direction and red line; polar direction) from **d**. **d**, High-resolution STM images of the 3 ML NaCl terrace ($5 \times 5 \text{ nm}^2$, $V_s = 200 \text{ mV}$, $I_t = 200 \text{ pA}$). Blue and red lines are the surface corrugations along non-polar and polar directions in **c**, respectively. Blue and blue-green spheres are superimposed to represent Na^+ ions and Cl^- ions, respectively.

[15] page 20:

- **Fig. 3 | Calculated energy profile of the lateral manipulation of a water molecule on 2 ML NaCl surface.**

[16] page 21:

- **Fig. 4 | Manipulation of the molecule with different heights.** **a**, Schematic of dragging adsorbate on a surface. Tip-adsorbate distance (r), total force (F_T), lateral component (F_L) and angle (ϕ) between F_T and F_L are given with a typical tip trace. **b**, The ϕ versus r plot. Blue and red points indicate that the molecule was manipulated along the non-polar and polar directions, respectively. Each point was averaged over at least six trials. The error bars indicate the standard deviation. The solid lines are the least-squares fitted lines of the data. As the tip approached the molecule, the required angle for manipulating the molecule increased to compensate the increased lateral component of the tip–molecule force. **c**, The $\Gamma (= r/\cos(\phi))$ versus r plot derived

from the fitted lines in **b**. Blue and red lines indicate that the molecule was manipulated along non-polar and polar directions, respectively.

[17] page 22:

- **Fig. 5 | Selective dissolution of a single Cl^- ion from the step.** **a,b**, STM images of water molecules at the step edge before and after manipulation along a white arrow, respectively ($5 \times 5 \text{ nm}^2$, $V_s = 300 \text{ mV}$, $I_t = 50 \text{ pA}$). **c**, High-resolution STM image of **b** showing a single Cl^- vacancy ($5 \times 5 \text{ nm}^2$, $V_s = 300 \text{ mV}$, $I_t = 500 \text{ pA}$). **d**, Flattened STM image of another example of the selective dissolution in **Supplementary Fig. 7j**, clarifying displaced ions near the vacancy ($5 \times 5 \text{ nm}^2$, $V_s = 400 \text{ mV}$, $I_t = 500 \text{ pA}$). In **c** and **d**, blue and blue-green dots indicate Na^+ and Cl^- ions and dotted circles represent vacant Cl^- sites of the steps. **e–h**, Side view of charge density difference and cross-sectional plots of the water molecule at the Na^+ site (**e,f**) and at the Cl^- site of the step (**g,h**) **calculated by DFT**. Blue, blue-green, red, and white spheres represent Na^+ ions, Cl^- ions, oxygen atoms and hydrogen atoms, respectively. In **e** and **g**, yellow (cyan) indicates an area of electron accumulation (electron depletion). In **f** and **j**, large (small) circle indicates the position of Cl^- ion (Na^+ ion).

*** In the revised supplementary information**

[1] page 1:

- Supplementary **Figs. 1 to 8**
- **Supplementary** References

[2] page 2:

- **Supplementary Fig. 1 | Quantitative analysis of the lateral manipulation.** **a**, Averaged amplitudes of tip height profiles. **b**, Averaged angles from tip height profiles. These data were derived from the manipulation profiles. Tunnelling current was 240 and 80 pA at positive and negative bias voltages, respectively. Each data was averaged over at least eight trials. The amplitude along the non-polar direction is lowered than that along the polar direction, while the angle along the non-polar direction is higher than that along the polar direction. **The error bars indicate the standard deviation.**

[3] page 3:

- **Supplementary Fig. 2 | Determination of the tip-Ag surface distance.**

[4] page 4:

- **Supplementary Fig. 3 | Effect of tip with various currents on lateral manipulation.**

a, Averaged amplitudes of tip height profiles measured from polar direction with various current at the bias voltage of 350 mV. **b**, Averaged angles obtained from tip height profiles. Each point was averaged over at least fifteen trials. The error bars indicate the standard deviation.

[5] page 5:

- **Supplementary Fig. 4 | Effect of tip on lateral manipulation.**

[6] page 6:

- **Supplementary Fig. 5 | Lateral manipulation of a single water molecule at a step.** **a**, Two water molecules at the step edge before manipulation ($2 \times 7 \text{ nm}^2$, $V_s = -350 \text{ mV}$ and $I_t = 50 \text{ pA}$). **b**, The water molecule moved along the arrow. The displacement was 0.564 nm . **c**, Representative optimized molecular configurations along the non-polar step calculated by DFT. **d**, Energy profile of the water molecule along the step edge.

[7] page 7:

- **Supplementary Fig. 6 | Recovery of Cl^- vacancy with the extracted anion.**

[8] page 8:

- **Supplementary Fig. 7 | Consecutive manipulations for selective dissolution in Fig. 5d.**

[9] page 9:

- **Supplementary Fig. 8 | Simulated STM images of perfect NaCl step edge and the step edge with different type of defects.** **a**, Simulated STM image of intact NaCl surface with two parallel step edges (top) and the corresponding atomic model (bottom). **b**, Simulated STM image of a Na vacancy on the step edge (top) and the corresponding atomic model (bottom). **c**, Simulated STM image of a Cl vacancy on the step edge (top) and the corresponding atomic model (bottom). **d**, Simulated STM image of a NaCl pair vacancy on the step edge (top) and the corresponding atomic model (bottom). In **c**, the symmetric structure of a single Cl vacancy matches the experimental STM image in Fig. 5d.

[10] page 10:

- lower the energy barrier¹.
- as a function of r (Supplementary Fig. 4a)
- using data in Fig. 4b
- derived from Morse potential².
- $$\frac{F_{\text{TH}}^{\text{P}}}{F_{\text{TH}}^{\text{NP}}} = \frac{F_{\text{TH}}^{\text{P},0} + k\Delta F}{F_{\text{TH}}^{\text{NP}} + \Delta F} = \frac{F_{\text{TH}}^{\text{P},0} + kF_{\text{TH}}^{\text{NP},0}\{\exp(ar + b) - 1\}}{F_{\text{TH}}^{\text{NP},0} \exp(ar + b)}$$

[11] page 11:

- We obtained $k = 1.0462$, $a = 0.00505$ and $b = -1.1630$ by applying this equation to Supplementary Fig. 4a.
- the energy barriers (Supplementary Fig. 4b,c).
- (state 3 and 19 in Supplementary Fig. 5)
- Cl^- ions using STM tip³⁻⁷.
- according to previous reports³⁻⁷.
- as demonstrated in the previous reports³⁻⁷.

[12] page 12:

- In the recent works by Jiang's group^{3,8}.
- feature at the Cl^- site (Fig. 5d)
- using DFT calculations (Supplementary Fig. 8)
- feature of the vacancy in Fig. 5d

[13] page 13:

Supplementary references

1. Emmrich, M. *et al.* Force field analysis suggests a lowering of diffusion barriers in atomic manipulation due to presence of STM tip. *Phys. Rev. Lett.* **114**, 146101 (2015).
2. Giessibl, F. J. Probing the nature of chemical bonds by atomic force microscopy. *Molecules* **26**, 4068 (2021).

3. Repp, J., Meyer, G., Paavilainen, S., Olsson, F. E. & Persson, M. Scanning tunneling spectroscopy of Cl vacancies in NaCl films: Strong electron-phonon coupling in double-barrier tunneling junctions. *Phys. Rev. Lett.* **95**, 225503 (2005).
4. Peng, J. *et al.* The effect of hydration number on the interfacial transport of sodium ions. *Nature* **557**, 701–705 (2018).
5. Schuler, B. *et al.* Effect of electron-phonon interaction on the formation of one-dimensional electronic states in coupled Cl vacancies. *Phys. Rev. B* **91**, 235443 (2015).
6. Li, Z. *et al.* Lateral manipulation of atomic vacancies in ultrathin insulating films. *ACS Nano* **9**, 5318–5325 (2015).
7. Meng, X. *et al.* Direct visualization of concerted proton tunnelling in a water nanocluster. *Nat. Phys.* **11**, 235-239 (2015)
8. Peng, J., Guo, J., Ma, R., Meng, X. & Jiang, Y. Atomic-scale imaging of the dissolution of NaCl islands by water at low temperature. *J. Phys.: Condens. Matter* **29**, 104001 (2017).

REVIEWERS' COMMENTS

Reviewer #1 (Remarks to the Author):

I am pleased with the revisions. I look forward to sharing this work with my students.

Reviewer #2 (Remarks to the Author):

Overall, we are satisfied with many of the changes made in response to our comments. As a reviewer, I raise issues with the expectation that changes will be made to the text, unless the authors disagree with my point. Here, the authors provide long exploratory discussions, in some cases without making changes to the text, which seems less useful to me.

Comment 1: In revised text, a paragraph with a discussion of the experiment versus dissolution taking place in a liquid is provided. That was a good addition.

However, authors did not add text to the manuscript discussing the energy scales we requested. The key question was: How do the energies calculated from DFT for the various possible configurations (and the resulting differences thereof) compare to the energies that set the scale for water's characteristics? The authors provided a discussion in the rebuttal, but did not modify the text, so that the relevant information is still not presented to the reader of the paper.

Comment 3: Figures throughout the paper were much improved. Revised Fig. 2c still lacks clarity. It would be useful for the authors to indicate the placement of chloride along the displacement axis, which could be done by overlaying transparent banding at the position of every chloride. It would also be useful to indicate the distinction between the top and bottom parts of 2c (for dragging water and tip only?).

Comment 5: Parts of the rebuttal are more clearly written than the manuscript on this subject, but it doesn't seem like any revisions to the text have been made that match. At the least, it would be useful for the authors to include this information in the SI.

Comment 13: changes to the text are insufficient, since the word 'reduced' was changed to 'closer', but the context is still not provided. What was the tip-molecule distance referred to on line 81?

Further comments on the paper as a whole:

It would be useful for the authors to carefully proofread the article for grammatical and stylistic errors. An example of the latter is to replace the last sentence by "The selective dissolution of the chlorine anion is attributed to its large polarizability."

Reviewer #3 (Remarks to the Author):

Evaluating manuscript's importance is more difficult than evaluating its correctness. I appreciate that single water molecule interaction with NaCl surface is studied experimentally. This is not an easy task. I find this study interesting and definitely publishable. I still feel that Phys. Chem. journal would be more appropriate, but I do not oppose the publication in Nature Comm. There are not that many atomistic level experiments in this area.

REPLY TO REVIEWERS' COMMENTS

Reviewer #1 (Remarks to the Author):

I am pleased with the revisions. I look forward to sharing this work with my students.

We deeply appreciate Reviewer 1's valuable comments on our works.

Reviewer #2 (Remarks to the Author):

Overall, we are satisfied with many of the changes made in response to our comments. As a reviewer, I raise issues with the expectation that changes will be made to the text, unless the authors disagree with my point. Here, the authors provide long exploratory discussions, in some cases without making changes to the text, which seems less useful to me.

We appreciate Reviewer 2's valuable comments and suggestions, which have greatly improved the quality of our manuscript.

[Comment 1]

In revised text, a paragraph with a discussion of the experiment versus dissolution taking place in a liquid is provided. That was a good addition. However, authors did not add text to the manuscript discussing the energy scales we requested. The key question was: How do the energies calculated from DFT for the various possible configurations (and the resulting differences thereof) compared to the energies that set the scale for water's characteristics? The authors provided a discussion in the rebuttal, but did not modify the text, so that the relevant information is still not presented to the reader of the paper.

Fig. R1 is the energy profile for dissolution of a Cl^- ion with a single water molecule at the step of NaCl, revealing that salt dissolution is an endothermic process. However, extending our model to include crowded water molecules would be a repetition of previous studies. Klimeš *et al.* reported the energy for dissolution of ions in various cases (Fig. R2) [ref. 23, *J. Phys. Chem.* **139**, 234702 (2013)]. They demonstrated that the displacement (dissolution) energy became negative for a Cl^- ion when many water molecules were involved in the dissolution process, which would be a good answer to Review 2's question. For this reason, we decide not to include Fig. R1 and the extended studies in the present manuscript.

Fig. R1 | Reaction energy profile and atomic structures for dissolution of a Cl^- ion at the step of NaCl surface. Blue, blue-green, red and white spheres are Na^+ ions, Cl^- ions, oxygen atoms and hydrogen atoms, respectively.

[redacted]

Fig. R2 | NaCl dissolution from a step. a, Defects on the NaCl surface with a monoatomic step. **b,** The displacement energy on a surface with a monoatomic step. Blue, blue-green, violet, green, red and white spheres are Na^+ ions, Cl^- ions, the extracted Na^+ ion, the extracted Cl^- ion, oxygen atoms and hydrogen atoms, respectively [J. Phys. Chem. **139**, 234702, (2013)].

[Comment 3]

Figures throughout the paper were much improved. Revised Fig. 2c still lacks clarity. It would be useful for the authors to indicate the placement of chloride along the displacement axis, which could be done by overlaying transparent banding at the position of every chloride. It

would also be useful to indicate the distinction between the top and bottom parts of 2c (for dragging water and tip only?).

The dips in the tip-height traces and the surface corrugations along both directions correspond to the position of Na^+ . We indicated the position of Na^+ in the revised Fig. 2, as Reviewer 2 commented.

[Comment 5]

Parts of the rebuttal are more clearly written than the manuscript on this subject, but it doesn't seem like any revisions to the text have been made that match. At the least, it would be useful for the authors to include this information in the SI.

In the previous rebuttal, we have explained the trace of the tip and an angle ϕ during the lateral manipulation. We did not include this explanation in the manuscript because this is one of the well-established analyses for interpreting the lateral manipulation of atom or molecules [Phys. Rev. Lett. **79**, 697-700 (1997) and Appl. Phys. Lett. **99**, 221902 (2011)]. The detailed explanation on the atomic and molecular manipulation can be easily found from many review articles and even from the book [Nanoelectronics and Information Technology, 3rd ed. Wiley-VCH (2012)]. For better clarity, we revised the manuscript as follows: “The periodic array of atoms of the substrate yields an equal displacement for each movement in the tip-height trace (Fig. 2e). A sudden jump in the trace implies that the water molecule is pulled by the tip and hops by one adsorption site towards the tip. The lateral motion of a water molecule is initiated when the lateral component (F_L) of the tip–molecule attractive force (F_T) exceeds the threshold frictional force between a molecule and a substrate (F_{TH})³¹. At the moment of jump, the lateral force reaches the threshold frictional force, and the angle (ϕ) between F_T and F_L can be obtained from the slope of the tip trace (Fig. 4a).”

[Comment 13]

The changes to the text are insufficient, since the word 'reduced' was changed to 'closer', but the context is still not provided. What was the tip-molecule distance referred to on line 81?

We revised the sentence as follows: “At a closer tip–molecule distance of about 300 pm, we moved the tip along the non-polar $\langle 100 \rangle$ and polar $\langle 110 \rangle$ directions on the terrace, as illustrated in Fig. 2a–d.”

Further comments on the paper as a whole:

It would be useful for the authors to carefully proofread the article for grammatical and stylistic errors. An example of the latter is to replace the last sentence by "The selective dissolution of the chlorine anion is attributed to its large polarizability.”

We revised the manuscript as Reviewer 2 commented.

Reviewer #3 (Remarks to the Author):

Evaluating manuscript's importance is more difficult than evaluating its correctness. I appreciate that single water molecule interaction with NaCl surface is studied experimentally. This is not an easy task. I find this study interesting and definitely publishable. I still feel that Phys. Chem. journal would be more appropriate, but I do not oppose the publication in Nature Comm. There are not that many atomistic level experiments in this area.

We deeply appreciate that Reviewer 3 finds our study interesting and definitely publishable.